# Adolescent Health Literacy in Beijing and Melbourne: A Cross-Cultural Comparison

**DOI:** 10.3390/ijerph17041242

**Published:** 2020-02-14

**Authors:** Shuaijun Guo, Xiaoming Yu, Elise Davis, Rebecca Armstrong, Elisha Riggs, Lucio Naccarella

**Affiliations:** 1Centre for Community Child Health, Murdoch Children’s Research Institute, Royal Children’s Hospital, Melbourne, VIC 3052, Australia; 2Melbourne School of Population and Global Health, University of Melbourne, Melbourne, VIC 3010, Australia; Elise.Davis@mindaustralia.org.au (E.D.); Rebecca.rmstrng@gmail.com (R.A.); l.naccarella@unimelb.edu.au (L.N.); 3Department of Paediatrics, University of Melbourne, Melbourne, VIC 3010, Australia; 4Institute of Child and Adolescent Health, School of Public Health, Peking University, Beijing 100191, China; 5Intergenerational Health Research Group, Murdoch Children’s Research Institute, Royal Children’s Hospital, Melbourne, VIC 3052, Australia; elisha.riggs@mcri.edu.au; 6Department of General Practice, University of Melbourne, Melbourne, VIC 3010, Australia

**Keywords:** health literacy, health outcomes, secondary students, cross-culture, China, Australia

## Abstract

While adolescent health literacy has gained momentum, it is under-researched from a cross-cultural perspective. This study aims to compare health literacy among two cultural groups of secondary students in Beijing and Melbourne. A cross-sectional study was conducted with 770 students from five secondary schools in Beijing and Melbourne. A self-administered questionnaire was designed to collect information on health literacy (the eight-item health literacy assessment tool (HLAT-8), the Newest Vital Sign (NVS) and the 47-item Health Literacy Survey (HLS-47)), its antecedents and health outcomes. Overall, students’ health literacy in Melbourne (n = 120) was higher than that in Beijing (n = 650): 28.25 ± 6.00 versus 26.37 ± 5.89 (HLAT-8); and 4.13 ± 1.73 versus 3.65 ± 1.64 (NVS). The proportion of students with low health literacy varied by instruments, representing 23.7–32.2% in Melbourne and 29.0%–45.5% in Beijing. In both cultural groups, students’ self-efficacy, social support, and perceptions of school environment were associated with their health literacy, which in turn predicted their health behaviours, patient-provider communication and health status. Given the nature of our study design and small samples, a cautious conclusion would be that adolescent health literacy is sensitive to the broad cultural context and might be an interactive outcome influenced by an individual’s health skills and the social environment. Particularly, creating a supportive school environment is critical to develop adolescent health literacy that would eventually contribute to better health outcomes.

## 1. Introduction

Over the last decade, the field of adolescent health literacy has gained momentum globally [1,2,3,4]. As a personal asset, it highlights the empowerment of adolescents and their own rights of citizenship in society [5]. Low health literacy in adolescents is associated with a range of adverse health outcomes including health-compromising behaviours, poor health status and overweight/obesity [6,7,8]. Adolescent health literacy is an important and modifiable determinant of health; promoting health literacy at an early age is a key intervention strategy to reduce disease burden and health disparities [9].

Compared to adult health literacy, adolescent health literacy is under-researched, particularly from a cultural and societal perspective [10]. The seminal health literacy framework proposed by the Institute of Medicine (IOM) highlights three settings (healthcare; education; culture and society) where health literacy can be developed and enhanced at both individual and population levels [11]. However, most current research focuses on health literacy in either the healthcare context or educational settings [12,13,14]. 

Cultural and social contexts shape an individual’s beliefs and languages and, therefore, influence health literacy [11]. The relationship between health literacy and an individual’s cultural beliefs and language backgrounds has been well documented, with immigrants and ethnic minority groups having a higher risk of low health literacy [10,15]. In the present study, we focused on the broad cultural and social environment, rather than an individual’s cultural beliefs and language backgrounds. Culture provides a context through which meaning is gained from health information, and provides the purpose by which people come to understand their health needs and take actions in healthcare, disease prevention and health promotion to maintain good health [11,16]. Similarly, the social environment (e.g., schools, families, communities) is well documented regarding the conditions over which an individual has little control but that affects his or her health literacy [1]. Understanding the role of the cultural and social environment in health literacy is important because this will inform strategic directions for health literacy interventions in this multi-cultural and globalized world, particularly in a diverse multicultural population. 

International health literacy studies revealed that the distribution of health literacy levels differed substantially across cultural groups [17,18,19]. For instance, the European Health Literacy Survey conducted in eight countries (Austria, Bulgaria, Germany, Greece, Ireland, the Netherlands, Poland and Spain) revealed that around half (47.6%) of respondents aged over 15 had low health literacy, with the prevalence ranging from 28.7% in the Netherlands to 62.1% in Bulgaria [17]. While these comparison studies provide an overall picture of health literacy between different cultural groups, they mostly focus on adult health literacy. The comparison of adolescent health literacy remains unclear. 

Using multiple measures of health literacy in a single study allows researchers to learn about how each measure performs, to compare results between different measures and to enhance the rigor of findings [20]. Pleasant et al. [21] recommended seven principles to advance the field of health literacy measurement. These principles include: (1) explicitly built on a conceptual framework; (2) multi-dimensionality in content and methodology; (3) measure health literacy on a continuous basis; (4) treat health literacy as a latent construct; (5) honour the principle of compatibility; (6) allow comparison across different contexts including across cultures; and (7) prioritize public health applications versus clinical screening. However, little is known about the utility of these principles in practice.

Given that health literacy is a broad concept with a wide range of assessment tools [3], it is essential to consider it within a specific context for a specific content [22]. In this present study, we defined adolescent health literacy as “an individual’s ability to find, understand and use health information and services to promote and maintain good health” [23] and applied it into school settings. Schools were chosen because they are critical venues for improving adolescent health literacy through school-based programs [13]. Additionally, schools are the most common places where adolescents spend most of their daytime. It was therefore feasible and achievable to recruit large samples in a short time. We targeted two cultural groups of school-aged adolescents in China and Australia due to two reasons. One reason is that low health literacy in adolescents is prevalent in both countries (93.7% in China [24]; 67.6% in Australia [25]). However, health literacy measurement is inequivalent and making its results incomparable. Using consistent measurement tools will enable researchers and policymakers to understand the status of health literacy across cultures and identify unique and/or common health literacy needs and influencing factors within each culture, thus contributing to directional strategies about next-step health literacy interventions. The second is an opportunistic reason because of the research team’s background and partnership networks in China and Australia.

Based on Pleasant’s health literacy measurement principles [21], we aimed to examine and compare health literacy between Chinese and Australian school-aged adolescents, in order to understand the effect of cultural and social environment on adolescent health literacy and identify areas for improvement across different cultural contexts. A secondary aim was to advance the practice of health literacy measurement.

## 2. Materials and Methods 

### 2.1. Study Population and Sampling Design

A cross-sectional study was designed to recruit adolescents from five secondary schools in two cities: four schools in Beijing, China, and one school in Melbourne, Australia, using cluster and convenience sampling. Utilizing existing partnerships with Beijing secondary schools, we selected two public government schools in a high socioeconomic district and two in a low socioeconomic district. At each school, two whole classes in each year level (Years 7, 8 and 9) were chosen, with the number of students in each class ranging from 20 to 35. In Melbourne, one public government school in high socioeconomic district was recruited and all students (n = 918) in Years 7–9 were invited to participate in the field survey. Ethics approval to conduct this study was obtained from The University of Melbourne (Ethics number: 1442884) and Peking University Institutional Review Board (Ethics number: IRB00001052-15024). To ensure the reporting and methodological quality of this study, the STROBE statement [26] and Pleasant’s health literacy measurement principles [21] were employed. Further details are presented in Appendix A and Appendix B.

### 2.2. Data Collection

The field survey in Beijing was conducted between November and December 2015. Passive and opt-out consent was obtained from both parents and students. Prior to data collection, the principal researcher gave a brief training to ten investigators (i.e., Master students in public health from Peking University) to ensure consistency of the administration. All secondary students were then asked to complete a self-administered print version questionnaire during class or a class break. 

The field survey in Melbourne was conducted between July and September 2016. Active and opt-in consent was obtained from both parents and students. With school representatives’ support, a web link of our questionnaire was sent to all students who had parental consent in Years 7–9 by class email. Students were invited to complete an online questionnaire when participating in the first health and physical education class in the third school term. 

### 2.3. Questionnaire

The English version questionnaire was developed first based on Manganello’s health literacy framework [1], which included students’ health literacy, key upstream factors (i.e., intrapersonal, interpersonal, and environmental factors) and health outcomes. Then it was translated into Chinese using a translation and back translation technique [27]. Given that some measurement scales (i.e., self-efficacy, social support, health literacy, health behaviours, patient-provider communication, and health status) have been translated and validated in Chinese adolescents, we only translated those without a Chinese version (e.g., school environment, community environment). 

#### 2.3.1. Intrapersonal Factors

Intrapersonal factors included socio-demographics and self-efficacy. Socio-demographics included age, gender (male or female), year level (Years 7, 8 or 9), family composition (living with two parents or other living arrangement), and family affluence level (low, medium or high) [28]. Personal self-efficacy was measured by the General Self-Efficacy Scale (GSES) [29], a 10-item scale that assessed personal belief in the ability to cope with a variety of challenges in life. Respondents indicated their level of agreement on a 4-point scale (1 = not at all true, 4 = exactly true). The GSES total score range was 10–40, with higher scores indicating higher levels of self-efficacy. In the present study, Cronbach’s α for the GSES was 0.89.

#### 2.3.2. Interpersonal Factors

Interpersonal factors were assessed using the Multidimensional Scale of Perceived Social Support (MSPSS) [30], a 12-item scale that measured an individual’s perceived support from family, friends and significant others. Respondents answered each item on a seven-point Likert scale (1 = very strongly disagree, 7 = very strongly agree). The MSPSS total score range was 12–84, with higher scores reflecting higher levels of social support. Cronbach’s α for the MSPSS was 0.93 for our sample.

#### 2.3.3. Environmental Factors

School environment was assessed by the School Environment Scale (SES), which was derived from the Communities That Care Youth Survey [31]. The SES comprised 10 items measuring students’ subjective feelings about opportunities and rewards for prosocial involvement at school. Respondents indicated their level of agreement with each statement on a four-point Likert scale (1 = strongly disagree, 4 = strongly agree). The SES total score range was 10–40, with higher scores suggesting stronger bonds of attachment to school. Cronbach’s α for the SES was 0.88 in this study. 

Community environment was assessed by the Community Environment Scale (CES), which measured respondents’ subjective feelings of their neighbourhood environment such as cleanliness and safety [32]. Participants answered each item on a five-point scale (1 = strongly disagree, 4 = strongly agree; 0 = do not know). The CES total score range was 0–36, with higher scores indicating a more liveable and supportive community. In this study, Cronbach’s α for the CES was 0.84.

#### 2.3.4. Health Literacy

Three health literacy instruments were used to compare results between different measures and enhance the rigor of findings [20]: the eight-item Health Literacy Assessment Tool (HLAT-8) [23], the six-item Newest Vital Sign (NVS) [33] and the 47-item Health Literacy Study-Asia-Questionnaire (HLS-47) [34]. The HLAT-8 and the HLS-47 were self-report instruments that measured an individual’s ability to access, understand, evaluate, and communicate health information in everyday life [23,34]; whereas the NVS was a performance-based measure for reading comprehension and numeracy [33]. The total score range was 0–37, 0–6 and 0–50, respectively, with higher scores indicating higher levels of health literacy. Scores of 4 to 6 for the NVS and scores of 33–50 for the HLS-47 indicated “adequate health literacy”. The NVS and the HLS-47 have shown satisfactory internal consistency and structural validity [34,35]. The HLAT-8 has been validated in Chinese secondary students [36], with a Cronbach’s alpha of 0.79. 

#### 2.3.5. Health Outcomes

Three health outcomes were assessed including health behaviours, health service use, and health status. Health behaviours were measured by five items (breakfast eating, physical activity, cigarette smoking, alcohol drinking and teeth brushing) derived from previously well-established student health and wellbeing surveys [37]. The total score for health behaviours is 5–35, with higher scores indicating more health-promoting behaviours. Health service use was assessed by a single item of patient-provider communication (‘How many times have you raised a question during your doctor’s appointment in the last 12 months?’ 1 = 0 time, 4 = 6 times or more) [34]. Health status was assessed using a widely-used general self-report health question (‘In general, would you say your health is?’ 1 = poor, 5 = excellent) [38].

### 2.4. Statistical Analysis

All statistical analyses were conducted using STATA 15.1 (StataCorp LLC, College Station, TX, USA). Descriptive statistics (frequency/percentage, mean, median) were first used to examine participants’ socio-demographics, both overall and by two locations. Univariate analysis (t-test, ANOVA, nonparametric test) was then conducted to examine the differences in health literacy, its antecedents and health outcomes between Beijing and Melbourne students. Finally, multivariate analysis (linear regression, logistic regression) was used to investigate the association between health literacy and its antecedents, health literacy and health outcomes. The individual mean substitution was conducted for non-response items in self-report scales. Data normality was assessed using skewness and kurtosis values. Results showed that scores on self-efficacy, health literacy and school environment were distributed normally, whereas scores on social support, community environment and health behaviours showed non-normal distribution.

## 3. Results

### 3.1. Participants’ Socio-Demographics

In total, 770 students in Years 7–9 were recruited from five secondary schools in Beijing (n = 650) and Melbourne (n = 120). The mean age of participants was 13.45 ± 1.02 (range: 11–17 years). The distribution of location, gender, year level, family composition and family affluence level are shown in Table 1.

### 3.2. Health Literacy, Its Antecedents and Health Outcomes

Table 2 shows differences in health literacy, its antecedents and each health outcome, both overall and by locations. Univariate analysis showed that students’ health literacy in Melbourne was higher than that in Beijing: 28.25 ± 6.00 versus 26.37 ± 5.89 (HLAT-8; *p* = 0.001); 4.31 ± 1.73 versus 3.65 ± 1.64 (NVS; *p* < 0.001). The proportion of students with low health literacy varied by self-report (HLS-47) and performance-based (NVS) instruments, representing 23.7%–32.2% in Melbourne and 29.0%–45.5% in Beijing. When examining the difference in health literacy antecedents, we found that Melbourne students had higher scores in self-efficacy (31.13 ± 5.08 versus 26.85 ± 6.37; *p* < 0.001) and community environment (29.68 ± 5.75 versus 25.89 ± 6.09; *p* < 0.001) than their counterparts in Beijing. In addition, there were also differences in scores of health behaviours and health status, with Melbourne students showing more health-promoting behaviours and better health status.

### 3.3. Association between Health Literacy and its Antecedents

Based on Manganello’s health literacy framework [1], we tested the relationship between health literacy and its antecedents according to each health literacy instrument. Here we laid emphasis on four modifiable factors that were potentially related to health literacy and reported their results in order to provide strategic implications for next-step intervention. After controlling for potential covariates (i.e., gender, year level, family composition, and family affluence level), we found consistent evidence on the positive association between overall health literacy and self-efficacy, social support, and school environment when using the HLAT-8 and HLS-47, respectively (Table 3). In addition, community environment was associated with student’s health literacy when using the HLS-47. On the contrary, the relationship between health literacy and its antecedents was not clear when using the NVS, but we observed a positive effect of self-efficacy, social support, school environment and community environment on health literacy (NVS). 

When comparing the standardized coefficients between independent variables, we found that school environment was the most significant influencing factor for health literacy, suggesting the important role of schools in developing students’ health literacy.

### 3.4. Association between Health Literacy and Health Outcomes

After controlling for all potential covariates (i.e., gender, year level, family composition, family affluence level, self-efficacy, social support, school environment, and community environment), we examined the relationship between health literacy and each health outcome according to each health literacy instrument respectively. Results showed that health literacy was positively associated with health-promoting behaviours (β = 0.06~0.07; *p* < 0.05), frequent patient-provider communication (OR = 1.03~1.07; *p* < 0.05) and good health status (OR = 1.03; *p* < 0.05) when using the HLAT-8 and the HLS-47. On the contrary, there was little evidence about the relationship between health literacy and all health outcomes when using the NVS (Table 4). 

## 4. Discussion

### 4.1. Summary of Key Findings

The present study investigated and compared secondary students’ health literacy in Beijing and Melbourne using different health literacy assessment tools. Specifically, there were three key findings: (1) differences in adolescent health literacy were observed between two cultural groups, with Melbourne students showing higher health literacy; (2) the use of different health literacy instruments resulted in different prevalence of low health literacy in both cultural groups; and (3) there was consistent evidence on the common determinants of health literacy and impacts of health literacy on health outcomes across both cultural groups. 

### 4.2. The Role of the Cultural Context in Predicting Adolescent Health Literacy

Consistent with previous theoretical and empirical studies [11,17], we found differences in health literacy between Melbourne students and Beijing students, suggesting adolescent health literacy is sensitive to the broad cultural context. Given that school health education is one of the main approaches to equipping adolescents with adequate health literacy skills [21], one possible explanation of our results is due to the cultural difference in the implementation of school health education between China and Australia. Although “health literacy” is explicitly included in the rationale and aims of school health education in both China and Australia [13,39], there are two key differences in terms of its conceptualization and the learning environment. Health literacy in China focuses on health knowledge and behaviours [39], whereas health literacy in Australia lays more emphasis on health skills such as critical thinking [13], which aligns with the component of health literacy measures (e.g., HLAT-8) used in the present study. In addition, China’s educational system traditionally expects students to be passive and not to question their teachers and health professionals [40]. This top-down pedagogical approach focuses on communication of health knowledge, rather than developing health skills and empowering students to consider how to take action to improve their health. Another likely reason is due to the fierce academic pressure in Chinese secondary schools, compared to Australian schools [41]. Due to limited class time of school health education and inadequate teaching resources, Chinese students are less likely to acquire enough health knowledge and skills for better healthy decision making in such an academic-focused culture [42]. Except for the above two reasons, there might be other social and structural determinants within each cultural group such as an individual’s beliefs, languages and attitudes [15]. To fully examine the impact of culture on health literacy at both the individual and societal level, there is a need for future research such as using qualitative methods to explore the underlying factors and mechanisms linking culture to adolescent health literacy.

### 4.3. Differing Performance of Health Literacy Measurement Tools

Compared with previous similar research in adults [43], the present study extends what is known about health literacy comparison in adolescents using different assessment tools. The NVS, a performance-based instrument [33], captured a higher proportion of students with low health literacy than the HLS-47, a self-report instrument [34]. One possible reason was the different underlying constructs of health literacy in each instrument. The NVS mainly focused on functional domain (reading and numeracy) [33], whereas the HLS-47 captured a more comprehensive health literacy which included functional, interactive (communication) and critical (evaluation and decision-making) domains [34]. Another possible reason was due to the measurement error inherent in self-report instruments. As shown in a study by Chew et al. [14], respondents with high self-efficacy were likely to rate highly in perceived competence with health. As the present study did not focus on the comparison of self-report and performance-based measures, whether this over-estimation exists needs to be explored in future research.

### 4.4. A Holistic Approach to Improving Adolescent Health Literacy

Consistent with existing theoretical frameworks [1,44], we found that adolescent health literacy was related to intrapersonal, interpersonal and environmental factors. Additionally, our findings are in line with previous empirical findings which suggested a close relationship between health literacy and personal self-efficacy [45], social support [46] and school environment [47]. Compared to previous quantitative studies [45,46,47], we did not analyse the relationship between health literacy and its antecedents separately. Instead, we considered all these antecedents together based on Manganello’s health literacy framework [1]. In addition, our finding extends the ecological evidence from a quantitative perspective using three different health literacy assessment tools. Understanding which influential factors are the most important can assist researchers to identify effective entry points for health literacy interventions. As shown in our results (standardized coefficients in Table 3), providing a supportive school environment seems to be the most effective intervention strategy in enhancing students’ health literacy. For example, the Health Promoting Schools (HPS) programs have been successful and widely accepted as a whole-school approach to provide students with a supportive environment to develop health literacy [48,49]. The key principles underlying these programs include clarifying a vision for health literacy in school policies, improving the school’s physical and social environment, and building partnerships with local communities [50,51]. 

Similarly, aligning with previous findings [6,52,53], we also found that adolescent health literacy was associated with a range of health outcomes, suggesting that promoting health literacy could be a useful strategy to improve student’s overall health. However, we did not verify the mediating role of health literacy in the relationship between its antecedents and each health outcome. Further research is needed to use longitudinal data and causal mediation analysis to examine the role of adolescent health literacy in predicting health outcomes.

### 4.5. Strengths and Limitations

We used two methodological frameworks to ensure the clarity, transparency, and rigor of the present study. First, Manganello’s health literacy framework was used to guide our study design and data collection [1]. Second, Pleasant’s health literacy measurement principles were employed as methodological guidelines to advance the practice of health literacy measurement [21]. Particularly, three different health literacy assessment tools were used to provide a comprehensive understanding of health literacy in secondary school students. 

Limitations should be noted. First, due to convenience sampling, students in Beijing and Melbourne secondary schools were not representative. Beijing and Melbourne secondary students’ health literacy might be higher than the general population of secondary school students because they live in metropolitan cities, which means that they have better access to education than their counterparts from disadvantaged backgrounds. Second, the sample size was not equivalent to Beijing and Melbourne secondary students. Due to recruitment challenges (e.g., active, opt-in consent from both parents and adolescents; busy educational commitment; no interest in research) and non-existing partnerships with Melbourne secondary schools, only 120 Australian students were recruited from one school. This may limit the statistical power of our findings for Melbourne students. Future research is needed to replicate our findings with larger and more representative samples across cultural groups. Third, this study only investigated the role of the broad cultural context in health literacy at the school level, rather than an individual’s cultural beliefs, ethnicities and language backgrounds. For example, we did not investigate the difference in Melbourne students’ health literacy by their country of birth and primary language spoken at home due to the small sample size. Further research is needed to explore the influence of culture at both the individual and societal level. Fourth, findings of the associations between health literacy, its influencing factors and health outcomes were based on cross-sectional data. Longitudinal studies or intervention studies with participant follow-up are needed to further confirm the causal relationship we observed here.

### 4.6. Implications

Given that health literacy measurement varies by its dimensions, health topics, forms of administration and participants’ characteristics [3], it is important to consider using Pleasant’s health literacy measurement principles as methodological guidelines to reduce disparities in health literacy measurement. For example, using multiple assessment tools can assist researchers to capture health literacy from different perspectives and verify if there are replicated findings, thus enhancing the rigor of evidence [21]. In the meantime, it remains unclear about the rationale of obtaining an accurate and continuous score of health literacy. Consistent with the approach used in previous studies [17,18,19], we treated each domain of health literacy measurement as having equal weight and then got an overall score. There is a need for future research to explore the degree to which each domain contributes to the overall health literacy score. Establishing such a specific scoring system will assist researchers to better understand the role of each domain and further advance the field of adolescent health literacy.

Despite the differing performance of each health literacy instrument, we found that at least one-third of adolescents did not have adequate health literacy skills, which in turn predicted their health outcomes. Compared to previous generations, adolescents nowadays face significant health challenges in the 21st century such as the shift in disease burden from traditional communicable diseases to non-communicable diseases and conditions [54] and challenges that adolescents face when using the Internet to access online health information [55]. Therefore, it is imperative to equip themselves with health literacy, particularly critical thinking and decision-making skills, to address these health challenges at an early age. In practice, low health literacy is not an individual issue, but an interactive outcome affected by an individual’s health skills and the broad cultural and social environment. Health literacy, as an outcome of school health education [48], should be continued and strengthened using the HPS framework to promote the health and well-being of all children and adolescents.

Culture is commonly associated with many antecedents of health disparities [56]. Our findings showed that cultural and social environments were related to adolescent health literacy. Cultural perspectives of health literacy help deepen the understanding of the global context within which health literacy plays an important role [57]. For example, health literacy initiatives such as the OPtimising HEealth LIterAcy (OPHELIA) toolkit can help address socio-cultural differences between health providers and patients (e.g., children and adolescents), thus narrowing the communication gap and improving access to health information and services [58]. Meanwhile, given that little empirical evidence is available from the young generation [10,59], there is also a need to examine the specific role of culture (e.g., ethnicity, race, ancestry) and how it interacts with health literacy and eventually contribute to health outcomes.

## 5. Conclusions

Adolescent health literacy is sensitive to the broad cultural context. While the prevalence of low health literacy varies when using different health literacy assessment tools, low health literacy is common, with at least one-third of adolescents facing challenges of accessing, understanding and using health information in everyday life. Except for the impact of culture on health literacy, there is consistent evidence showing that adolescent health literacy is associated with personal self-efficacy, social support, and perceptions of school environment in both cultural groups. In addition, adolescent health literacy is related to health behaviours, patient-provider communication, and health status. Given the nature of our study design and small samples, we have made a cautious conclusion that adolescent health literacy might be an interactive outcome influenced by an individual’s health skills and the social environment. Particularly, creating a supportive school environment is critical to develop adolescent health literacy that would eventually contribute to better health outcomes.

## Figures and Tables

**Table 1 ijerph-17-01242-t001:** Socio-demographics of study participants.

Participant Characteristics	Mean ± SD or Frequency (%)			
	Total	Beijing	Melbourne	Statistic (*p*)
Age	13.45 ± 1.02	13.42 ± 1.01	13.63 ± 1.03	−2.15 (0.032)
Location				
Beijing	650 (84.4)	-	-	-
Melbourne	120 (15.6)	-	-	
Gender				
Male	430 (55.8)	357 (54.9)	73 (60.8)	1.44 (0.231)
Female	340 (44.2)	293 (45.1)	47 (39.2)	
Year level				
Year 7	264 (34.3)	232 (35.7)	32 (26.7)	7.98 (0.018)
Year 8	250 (32.5)	215 (33.1)	35 (29.2)	
Year 9	256 (33.3)	203 (31.2)	53 (44.2)	
Family composition				
Living with two parents	692 (90.0)	572 (88.1)	120 (100.0)	15.82 (<0.001)
Other living arrangement *	77 (10.0)	77 (11.9)	0	
Family affluence level				
Low	181 (23.7)	179 (27.8)	2 (1.7)	112.30 (<0.001)
Medium	324 (42.4)	296 (46.0)	28 (23.3)	
High	259 (33.9)	169 (26.2)	90 (75.0)	

SD, Standard Deviation. ***** Other living arrangement included living with a single parent or living with a relative, or living in a shared care institution.

**Table 2 ijerph-17-01242-t002:** Distribution and univariate analysis of health literacy, its antecedents and health outcomes.

Key Variables	Mean ± SD or Frequency (%)			
	Total	Beijing	Melbourne	Statistic (*p*)
**Health literacy**				
HLAT-8	26.66 ± 5.94	26.37 ± 5.89	28.25 ± 6.00	−3.21 (0.001)
NVS	3.74 ± 1.64	3.64 ± 1.64	4.31 ± 1.52	−4.05 (<0.001)
Low health literacy (0–3 scores)	321 (43.0)	288 (45.5)	33 (29.2)	10.38 (0.001)
High health literacy (4–6 scores)	425 (57.0)	345 (54.5)	80 (70.8)	
HLS-47	36.95 ± 9.41	36.80 ± 9.59	37.72 ± 8.40	−0.97 (0.332)
Low health literacy (0–33 scores)	203 (28.2)	175 (29.0)	28 (23.7)	1.37 (0.242)
High health literacy (34–50 scores)	518 (71.8)	428 (71.0)	90 (76.3)	
**Antecedents**				
Self-efficacy	27.52 ± 6.37	26.85 ± 6.37	31.13 ± 5.08	−6.94 (<0.001)
Social support	63.21 ± 15.03	62.79 ± 15.26	65.49 ± 13.51	−1.50 (0.133)
School environment	30.41 ± 5.44	30.48 ± 5.59	30.00 ± 4.53	0.89 (0.375)
Community environment	26.49 ± 6.19	25.89 ± 6.09	29.68 ± 5.75	−7.63 (<0.001)
**Health outcomes**				
Health behaviours	30.21 ± 3.56	30.07 ± 3.61	30.95 ± 3.20	−2.40 (0.017)
Patient-provider communication				
No (0 time)	383 (51.8)	332 (53.2)	51 (44.0)	3.34 (0.067)
Yes (at least 1 time)	357 (48.2)	292 (46.8)	65 (56.0)	
Health status				
Fair or poor	241 (31.3)	224 (34.5)	17 (14.2)	19.40 (<0.001)
Excellent or very good or good	529 (68.7)	426 (65.5)	103 (85.8)	

HLAT, the 8-item Health Literacy Assessment Tool; HLS-47, the 47-item Health Literacy Survey; NVS, the Newest Vital Sign; SD, Standard Deviation.

**Table 3 ijerph-17-01242-t003:** Multiple linear regression models predicting health literacy according to different instruments.

Independent Variables	Health Literacy (HLAT-8) ^#^						Health Literacy (NVS) ^#^						Health Literacy (HLS-47) ^#^					
	Total *		Beijing		Melbourne		Total *		Beijing		Melbourne		Total *		Beijing		Melbourne	
	Beta	95% CI	Beta	95% CI	Beta	95% CI	Beta	95% CI	Beta	95% CI	Beta	95% CI	Beta	95% CI	Beta	95% CI	Beta	95% CI
Self-efficacy	**0.12**	**0.04, 0.18**	**0.12**	**0.03, 0.18**	0.16	−0.12, 0.48	0.02	−0.02, 0.03	0.02	−0.02, 0.03	0.07	−0.06, 0.10	**0.12**	**0.05, 0.29**	**0.10**	**0.02, 0.27**	0.19	−0.04, 0.68
Social support	**0.19**	**0.04, 0.10**	**0.19**	**0.04, 0.11**	0.18	−0.03, 0.18	0.07	−0.00, 0.02	0.07	−0.00, 0.02	0.09	−0.02, 0.04	**0.17**	**0.05, 0.16**	**0.16**	**0.04, 0.15**	**0.25**	**0.03, 0.28**
School environment	**0.23**	**0.15, 0.34**	**0.25**	**0.16, 0.36**	0.07	−0.24, 0.42	0.05	−0.01, 0.04	0.06	−0.01, 0.05	−0.09	−0.13, 0.06	**0.20**	**0.20, 0.51**	**0.21**	**0.20, 0.53**	0.20	−0.03, 0.78
Community environment	0.03	−0.04, 0.10	0.03	−0.04, 0.10	0.02	−0.16, 0.21	0.07	−0.00, 0.04	0.08	−0.00, 0.05	0.00	−0.05, 0.06	**0.11**	**0.05, 0.27**	0.08	−0.00, 0.25	**0.23**	**0.11, 0.56**

Beta, standardized coefficient; CI, Confidence Interval of unstandardized coefficient; HLAT, the eight-item Health Literacy Assessment Tool; HLS-47, the 47-item Health Literacy Survey; NVS, the Newest Vital Sign. ^#^ Health literacy was measured as a continuous dependent variable; ***** Location was additionally added as a controlling variable. All models were conducted, controlling for gender, year level, family composition and family affluence level. Bold values are those with *p* value less than 0.05.

**Table 4 ijerph-17-01242-t004:** Multivariate regression models predicting health outcomes according to different instruments.

Independent Variable	Health Behaviours ^#^						Patient-Provider Communication ^&^						Health Status ^&^					
	Total *		Beijing		Melbourne		Total *		Beijing		Melbourne		Total *		Beijing		Melbourne	
	β	95% CI	β	95% CI	β	95% CI	OR	95% CI	OR	95% CI	OR	95% CI	OR	95% CI	OR	95% CI	OR	95% CI
Health literacy																		
HLAT-8	**0.06**	**0.01, 0.10**	**0.06**	**0.00, 0.11**	0.08	−0.02, 0.17	**1.07**	**1.04, 1.11**	**1.07**	**1.03, 1.10**	**1.10**	**1.01, 1.20**	1.03	1.00, 1.06	1.03	0.99, 1.06	1.08	0.98, 1.19
NVS	0.06	−0.09, 0.20	0.10	−0.07, 0.26	−0.19	−0.50, 0.11	1.04	0.95, 1.15	1.08	0.98, 1.19	0.78	0.58, 1.05	1.00	0.90, 1.11	1.01	0.91, 1.12	0.82	0.51, 1.29
HLS-47	**0.07**	**0.04, 0.10**	**0.06**	**0.02, 0.09**	**0.15**	**0.08, 0.23**	**1.03**	**1.01, 1.05**	**1.03**	**1.01, 1.05**	**1.08**	**1.01, 1.15**	**1.03**	**1.01, 1.05**	**1.03**	**1.01, 1.05**	1.05	0.96, 1.15

β: unstandardized coefficient; CI, Confidence Interval; HLAT, the eight-item Health Literacy Assessment Tool; HLS-47, the 47-item Health Literacy Survey; NVS, the Newest Vital Sign; OR, Odds Ratio. ^#^ Separate multiple linear regression models were conducted to predict health behaviours according to each health literacy instrument; ^&^ Separate multiple logistic regression models were conducted to predict patient-provider communication and health status according to each health literacy instrument; * Location was additionally added as a controlling variable. All models were conducted, controlling for gender, year level, family composition, family affluence level, self-efficacy, social support, school environment and community environment. Bold values are those with *p* value less than 0.05.

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
