# Peer review of "Adolescent Health Literacy in Beijing and Melbourne: A Cross-Cultural Comparison"

_ijerph, 2020, doi:10.3390/ijerph17041242_

Round 1

Reviewer 1 Report

As a researcher in the area of adolescent health literacy, I am always glad to see other researchers involved. However, I was very disappointed with this study. 

Let me say first that you did a meticulous job of presenting your findings. Your work was well organized and  easy to understand. You mostly presented appropriate citations, and your work is innovative in looking across cultures and countries.

Unfortunately, I cannot support publication of this manuscript. I was very concerned to see the difference in participant pools between China and Australia. In my opinion, these differences are not merely a 'limitation' but rather a fundamental problem that negates any conclusions. Not only do you have a problem with power given the small number of Australian students, but also all of them were from a high SES group. The large group of Chinese students were a mix of high and low SES. This difference makes comparison inadvisable at best. I'd suggest redoing the study to include low SES Australian students which would also increase numbers.

More minor improvements include a citation for your claim that adolescent health literacy is under researched. I'm also somewhat disturbed by your use of 'intact family'. Surely in 2020 we are no longer categorizing families as those with biological parents. What about gay/lesbian couples? Step parents? And so on. I'd advise you to drop this outdated concept.

Given the novelty of your work, I hope you will return to the field and rectify some of these areas of weakness. I would be pleased to review a revised manuscript.

Reviewer 2 Report

This paper reports on a cross-sectional study was conducted with students from secondary schools in Beijing and Melbourne and compares health literacy, antecedents to health literacy and health outcomes between Australian and Chinese students. This manuscript is well-written, includes a thorough description of the literature and theoretical framework with relevant citations, and describes clear and transparent methods.

This reviewer suggest some major and minor edits described below.

Major;

The rationale for comparing Australian and Chinese students is unclear. Study authors explain that they are comparing cultures and cite European studies. But it is not evident why they are comparing these two cultures? And how this comparison moves the field forward? There is Limited description of the bi-variate and univariate results; limited description of the results in Table 3 and 4. Please expand the written results section for tables 3 and 4. In the methods, the author describes univariate analysis (t-test, ANOVA, nonparametric test). Where are these results described in the paper? The overall take-home message is missing. Why are these findings important for the field, future work or programs? Please expand on these ideas in the Conclusion and please include a more compelling conclusion in the abstract.

Minor edits:

Tables – Improve table titles to include the specific analyses displayed. Specifically, what analyses are presented in Tables 3 and 4? Please expand the footnote to detail analyses, survey wording and co-variates, if multivariate. Add a line to separate the different analyses in Tables 3 and 4 (for instance between HLAT and NVS analyses). It is hard to read these columns.

Reviewer 3 Report

The manuscript is well written and the rationale for the study is well argued. The results are very interesting, especially as potential antecedents were controlled for. This is a good manuscript.

The introduction does not review the rich literature on HL and outcomes. There are good reviews on it.

Methods - Reference to the HL measures should be provided next to the measure in the “health literacy” section under “data collection”.

The structure of the methods section is unclear. The measures appear under the heading of “data collection”. Not enough details are provided (references, reliabilities).

Results - The multivariate analyses are not well presented in the text. The text does not report controlling for background variables and this is only reported under the table (#3). For example, Table 3 (whose title is “association between…) seems to report univariate analyses between 2 variables, e.g. HL and self-efficacy, yet it is obviously a multivariate analysis as Beta is reported and a model is alluded to in explanations under the table. The authors do not explain – conceptually -  why they examine association between antecedents of health literacy  while controlling for some other antecedents (affluence level and others). Such an analysis requires a theoretical/conceptual explanation. The authors should also dwell on the results in Table 3 and not refer to them only in the discussion.

Discussion - Pls refrain from using words of cause and effect (e.g., words such as “affected”) in the discussion; the design is cross-sectional and not experimental or even longitudinal.

The discussion should view the results of the current study in relation to previous findings on antecedents and outcomes of HL.

Round 2

Reviewer 1 Report

While I see that you have added 'cautions' to your manuscript, I am still not entirely pleased with this work. Nevertheless, I trust that readers will take into account the difficulties with methodology and conclusions.